# It helps to talk: A guiding framework (TRUST) for peer support in delivering mental health care for adolescents living with HIV

Carol Wogrin[1]*, Nicola Willis[1], Abigail Mutsinze[1], Silindweyinkosi Chinoda[2], Ruth Verhey[2], Dixon Chibanda[2,3,4], Sarah Bernays[5,6]

**1** Africaid, Avondale, Harare, Zimbabwe, **2** Friendship Bench, Avondale, Harare, Zimbabwe, **3** Department of Psychiatry, University of Zimbabwe College of Health Sciences, Harare, Zimbabwe, **4** Centre for Global Mental Health, London School of Hygiene and Tropical Medicine, London, United Kingdom, **5** Global Health and Development, London School of Hygiene and Tropical Medicine, London, United Kingdom, **6** School of Public Health, University of Sydney, Sydney, Australia

* carol@zvandiri.org

## Abstract

### Introduction

Adolescents living with HIV have poor treatment outcomes, including lower rates of viral suppression, than other age groups. Emerging evidence suggests a connection between improved mental health and increased adherence. Strengthening the focus on mental health could support increased rates of viral suppression. In sub-Saharan Africa clinical services for mental health care are extremely limited. Additional mechanisms are required to address the unmet mental health needs of this group. We consider the role that community-based peer supporters, a cadre operating at scale with adolescents, could play in the provision of lay-support for mental health.

### Methods

We conducted qualitative research to explore the experiences of peer supporters involved in delivering a peer-led mental health intervention in Zimbabwe as part of a randomized control trial (Zvandiri-Friendship Bench trial). We conducted 2 focus group discussions towards the end of the trial with 20 peer supporters (aged 18–24) from across 10 intervention districts and audio recorded 200 of the peer supporters' monthly case reviews. These data were thematically analysed to explore how peer supporters reflect on what was required of them given the problems that clients raised and what they themselves needed in delivering mental health support.

### Results

A primary strength of the peer support model, reflected across the datasets, is that it enables adolescents to openly discuss their problems with peer supporters, confident that there is reciprocal trust and understanding derived from the similarity in their lived experiences with HIV. There are potential risks for peer supporters, including being overwhelmed by

**Data Availability Statement:** Data can not be publicly shared due to it being interviews with minors and reveals their HIV status and potentially

the HIV status of others, including those in their households. The data cannot be effectively anonymised without losing valuable contextual details necessary for conducting appropriately detailed analyses. Data is stored on Africaid's Figshare. Upon reasonable request, data can be accessed through contacting Vivian Chitiyo, Knowledge and Information Officer, Africaid, 11-12 Stoneridge Way North, Avondale, Harare, Zimbabwe; info@africaid-zvandiri.org; phone: +263 4 335 805.

**Funding:** Africaid (NW, AM, CW) and Friendship Bench (DC, RV, SC); grant number: G-1710-02137; Children's Investment Fund Foundation; ciff.org; No, the funders had no role in study design, data collection and analysis, decision to publish, or preparation of the manuscript.

**Competing interests:** The authors have declared that no competing interests exist.

engaging with and feeling responsible for resolving relationally and structurally complex problems, which warrant considerable supervision. To support this cadre critical elements are needed: a clearly defined scope for the manageable provision of mental health support; a strong triage and referral system for complex cases; mechanisms to support the inclusion of caregivers; and sustained investment in training and ongoing supervision.

## Conclusion

Extending peer support to explicitly include a focus on mental health has enormous potential. From this empirical study we have developed a framework of core considerations and principles (the TRUST Framework) to guide the implementation of adequate supportive infrastructure in place to enhance the opportunities and mitigate risks.

## Introduction

Adolescents living with HIV (ALHIV) have worse outcomes in terms of treatment, viral load suppression and mortality than do HIV positive children and adults. In sub-Saharan Africa, where 80% of the global total of 1.74 million ALHIV reside [1], they are the only group for whom mortality rate is increasing [2]. The evidence indicates that across chronic illnesses adolescents find it more difficult to sustainably comply with their treatment regimens than do people of other ages [3]. Adherence to HIV treatment is further complicated by the many adverse social consequences of having an HIV positive status [4, 5].

There is also an emerging recognition of the substantial unmet mental health needs of ALHIV, who have higher rates of depression and other common mental health problems than do their HIV negative peers [6, 7]. Although still an emerging area of enquiry, this is considered to be due to both the burden of HIV management and the higher likelihood that an adolescent living with HIV has experienced serious loss in various forms. In Sub-Saharan Africa where the burden of adolescent HIV is concentrated, a fifth of children in southern Africa are orphans, predominantly due to HIV [8]. There are frequently associated losses following parental death including household mobility, separation from siblings, loss of belongings, and disrupted school attendance, compounded by extreme poverty and pervasive stigma and discrimination [9], all of which are factors that contribute to the development of psychological problems [10]. Increasingly there is a connection being made between mental health and well-being with adherence and viral load suppression [4, 11–13]. Although the understanding of the relationships between mental health and HIV outcomes is still evolving, there is an increased recognition of the potential effect of attending to mental health as an important outcome in itself, but also tackling the psychological problems that play a role in poor adherence in ALHIV as a means to progress towards achieving the ambitious global target of 95% of those on Anti-retroviral therapy (ART) to be virally supressed [14–16].

In Zimbabwe and throughout the region, clinical services for mental health care are extremely limited, due to a lack of human resources and of service delivery programs, with fewer than 1 psychiatrist per 1,000,000 population and a similarly small number of psychologists [17]. Child and adolescent psychiatrists are all but non-existent. Scarce available care is concentrated in urban settings. In rural settings mental health care is largely delivered by a very small cohort of mental health nurses [17]. Given the lack of existing mental health resources and the urgency and scale of the unmet need, we need to look for alternative mechanisms through which the problem can be addressed [14, 18].

Peer support, which has been shown to have a promising impact on viral suppression and is being delivered at scale [18], may be one potential pathway. A recent trial of the Zvandiri peer support model, which uses differentiated service delivery, showed an increase of viral load suppression among ALHIV of 42% compared to standard care [19]. Despite enthusiasm and donor funding for peer support, its impact on beneficiaries and on the peer supporters themselves has received relatively little empirical attention [20–22]. Also, to date peer support among ALHIV has focused primarily on prevention and general psychosocial support and has not delved into addressing more complex issues associated with common mental health disorders.

In this paper we present the qualitative findings from the Zvandiri–Friendship Bench study, a mixed methods study including a cluster randomised controlled trial, examining the effectiveness of peer-delivery of mental health care on viral suppression and mental health among ALHIV over a period of 12 months, utilizing a problem-solving therapy intervention (Friendship Bench) [23]. Friendship Bench is a community based psychological support organization where lay cadres implement Problem Solving Therapy (PST). The model has been shown to be effective among adults [11, 24–26], but this is the first time that it has been adapted to be delivered by young people for ALHIV, with the delivery done by Zvandiri's Community Adolescent Treatment Supporters (CATS). We explore the perspectives and needs of young peer supporters as described by themselves when they take on the complexities of expanding their role to addressing the difficult life experiences that challenge the mental health of their peers. Based on our analysis of their accounts, we develop a framework outlining the overarching support and training principles (TRUST) that should be provided to guide the expansion of the peer supporter's role in delivering critically needed mental health care to ALHIV.

## Methods

The study was conducted in rural districts in Zimbabwe in 2018–2019. Sixty clinics within 10 districts were randomised 1:1 to either the intervention or control arm. ALHIV attending the control arm clinics received standard CATS support. Those attending the intervention arm clinics received support from CATS with additional training from Friendship Bench. The intervention consisted of standard CATS support plus a series of individual sessions focused on identifying the problems a young person was facing that were making their life difficult and exploring the support needed and strategies they could employ to improve their coping and resilience [23].

The CATS were recruited using standard Zvandiri selection criteria, including: aged 18–23, HIV positive, virally suppressed, able to read and write, psychologically stable, and able to serve as a role model. All CATS had received the standard initial 5-day CATS training focused on understanding HIV and its treatment, including psychosocial aspects of living with HIV, basic counselling techniques and support strategies enabling them to work within Ministry of Health and Child Care facilities providing adherence support and outreach to children and young people living with HIV in their communities. In addition, CATS delivering the intervention received a 5-day training from Friendship Bench. There were 2 PST trained CATS in each of the 30 intervention facilities. Trial participants, ALHIV aged 10–19 were recruited from the ART registers and screened for common mental disorder using the SSQ-14 [23].

In order to better understand the process of the CATS' implementing Friendship Bench's PST model to address depression and anxiety in their peers, we collected qualitative data to capture concurrent and retrospective accounts of the CATS' experiences of delivering mental health support to clients (Table 1). Case reviews (CR) between 20 individual CATS and their

**Table 1. Qualitative data collection description.**

| Method | Outline of method | Purpose of data collection | Timing | Sample | Total data collection points |
|---|---|---|---|---|---|
| Case Reviews | Individual supervisory discussions (between CATS and mentors) about client cases | CATS' experiences of provision of support and support needs in real-time | Each month over course of trial (Jan-Dec 2019) | 2 CATS in 10 intervention districts (n = 20) | N = 200 (feasibility constrained completion each month) |
| Focus Group Discussions | Group discussion between CATS | Retrospective reflections of CATS' on their experiences | End of trial (January 2020) | 20 CATS (10/ group) | N = 2 |

mentors were conducted each month, when feasible, over the 12-month trial duration. Two focus group discussions (FGD) were held with 20 CATS at the end of the trial. Conducted by Zvandiri researchers at the Zvandiri offices in Harare, the FGDs lasted approximately 90 minutes and involved a range of activities to facilitate reflective discussions about their experiences.

The case reviews and FGDs were conducted in the local languages and audio recorded. They were transcribed and translated into English. The CR transcripts generally ranged from 500–1000 words. A thematic analytical approach was adopted [27]. After extensive reading of the datasets, the FGDs and approximately 20 CRs were open coded by CW, AM, SC and SB. A coding framework was developed which was then applied to the remaining data. The datasets were first analysed separately and then comparatively attending to any differences between them. Analytical memos and weekly analytical team meetings were also used in the development of themes and identification of patterns related to CATS experiences across the datasets [28]. To maximise the value of this thematic analysis, we organized the key implications from our analysis into a framework of guiding operational principles to ensure that CATS are adequately supported to conduct mental health support on a sustained basis.

Ethics approval was given by the Medical Research Council of Zimbabwe #A/2324. The CATS provided written informed consent.

## Results

In the CATS' accounts of supporting their peers who were dealing with depression and anxiety, participants described a number of challenges in their experiences. We identified three overarching themes. These were: the process, described as the journey, of learning to talk about mental health problems; support structures and strategies for the CATS to deliver sustained care; and the critical role of support and supervision. The journey, learning to talk about mental health problems further broke down into four sub-themes: strengths of peer supporters, challenges in 'moving into mental health support, the complexity and relational embeddedness of the problems faced by ALHIV, and making it 'less bad': revised definitions of success.

We present the findings from both the FGDs and CRs together, as the main themes identified were consistent across the two datasets. In our presentation of data extracts we note the different sources and districts they come from.

### The journey: Learning to talk about mental health problems

**Strengths of peer supporters.** Peer supporters' lived experiences and training provide them with the knowledge they need to communicate an understanding of what clients are going through, offer a trusted ear and present as role models for coping with and managing their HIV. The CATS considered themselves to be uniquely positioned to provide effective support to ALHIV having experienced adolescence with an HIV positive status and being

close in age. They know the demands of daily pills, adherence challenges and the common desire for secrecy to avoid stigma and discrimination. Ordinarily this need for secrecy, which can be so disruptive to ALHIV's relationships and wellbeing, also robs them of accessing social support which could help them cope with difficult situations. However, with peer supporters, with whom they share the same status, they are more able to talk openly; "Some issues they cannot discuss with friends but are free to discuss with CATS. One girl, who had a broken relationship, could not share it with her mother because she would not like it." (CR, Kwekwe)

**Challenges in 'moving into' mental health peer-support.** The CATS described how it took time to earn the client's confidence. Having learnt through experience that their HIV status could become the subject of gossip; they did not automatically trust in the absolute confidentiality of peer support. Despite their connection through age and status, the youth of the CATS meant that the clients initially doubted they would have the wisdom or influence to do anything about their problems. A CATS explained, "They will be telling themselves that if I tell Tatenda my problem, how can she help me since she is just a CATS." (CR, Matobo)

This lack of confidence was shared by the CATS themselves, who expressed reticence and concern around their nascent competence when engaging in this new terrain. The similarity of their shared painful life experiences, while a potential strength in terms of understanding, also posed some challenges for the CATS, as many had not yet learnt how to talk about and to mediate the harm of their own problems. They are products of the same communities as the clients, where young people have limited opportunity to talk about their relational and contextual problems. They, too, are unlikely to have yet developed a language to apply to their own experience or the scaffolding for their resilience. Talking with clients about their problems helped CATS reflect on their own experiences, "As a CATS l managed to know myself and can now manage myself." (FGD, Hwange)

It is feasible that many of the CATS may not have reflected on and received support for their own challenges and losses and so it may be difficult for them to move into a 'role model' position, as they were more readily able to do with HIV specific issues such as adherence management and acceptance of their status. This also predisposed them to be triggered when engaging with similar experiences, and for which they needed additional support through discussion of these issues with their supervisors and mentors. Explained by one CATS, "In some cases as CATS you deal with an issue that once personally affected you. I think there is need for us as CATS to be supported like what we do with the participant." (CR, Kwekwe)

**Problems faced are complex and relationally embedded.** The problems clients spoke about with CATS included being orphaned, feeling poorly treated by caregivers and discriminated against at home and structural conditions related to poverty. These complex, relational problems were embedded within the social world of the adolescent, as opposed to being primarily intrapsychic. They were problems over which the ALHIV, both the clients and the CATS, had limited control due to their age and developmental level and required the intervention of adults in positions of authority who surrounded the adolescent. A CATS reflects on a problem commonly faced by clients of not having the fees to attend school, "He [client] told them [grandmother and maternal aunties] but they did not take any action. He continued to be sent home (from school due to no fees)." (CR, Gokwe South)

Having aided them to articulate the problem, the CATS often attempted to help clients to address it. If they could not, in part due to having insufficient influence on the situation, this risked leaving both clients and CATS feeling overwhelmed and helpless:

Her grandmother insulted her over the death of her parents. She became depressed to such an extent that she doesn't take her medication. She wanted her grandmother's friend to talk

to her grandmother. The plan did not work out because the friend refused to talk to her grandmother claiming that she wouldn't understand her. (CR, Murewa)

Accompanying their commitment to their role, the CATS described an acute sense of responsibility to 'fix' clients' problems they commonly encountered and felt helpless against, whether due to lack of referral sources or resources, "They expect me to solve their problems or maybe give them that thing that they are in need of like money." (CR, Zaka) CATS talked about the challenges of carrying clients' concerns around with them. This was a potential threat to their own mental health and points to their need for both training and supervision.

**I can make it less bad: Revised definitions of success.** Despite these challenges, all the CATS considered that providing the opportunity to be heard and understood was helpful to their clients. While they described some problems as being resolvable, there was an evolving understanding that for others amelioration, or harm reduction, was still a successful outcome. For example, "The community used to discriminate her, and through our sessions she understood that people are always saying something, good or bad she will take her medication." (CR, Kwekwe) Another example was a child who was happy with a 'successful' intervention when aggressive bullying was stopped, even though his hope of gaining social acceptance had not been achieved.

However, this realisation took time and the CATS highlighted that they wanted help to accelerate their own recalibration of what they can realistically expect to achieve in their roles.

## Support structures and strategies for the CATS to deliver sustained care

Some problems required the engagement of formal services through government and non-government organizations, including social services and those providing access to resources, "He needed assistance to get his birth certificate so that he could be able to get piece jobs." (CR, Chivi) The CATS described how much easier their roles became when they understood when and how to initiate referrals to these external systems. This enabled the CATS to connect clients with organisations and actors with greater agency to influence situations and, where appropriate, to accompany them through the process:

> On the issue of abuse by her stepfather, we reported the issue to Social Welfare. It is being handled. They are trying to see how they can remove her from home and place her in a much safer place. (CR, Chivi)

Despite their developing autonomy, adolescents' daily lives still tend to be heavily influenced by their relationships with their caregivers and household members. We found that there were critical opportunities for caregivers, who acted as influential gatekeepers, to be brought into the process to strengthen the provision of peer-delivered mental health support.

The CATS described the importance of pre-engagement activities with caregivers to mitigate initial scepticism or resistance. In some instances it took time and persistence for the CATS to win caregivers' trust and approval. The CATS felt that when adequate groundwork was laid by their mentors, they didn't have to struggle to overcome the scepticism of caregivers and had a much easier time entering into the work with their clients:

> When caregivers support groups were held caregivers accepted what was happening with their children. Caregivers were told that their children were going to be working with CATS and they accepted it. (FGD, Kwekwe)

The CATS emphasised the value of establishing a constructive rapport with caregivers in enabling a more receptive response to any intervention, "The parent is the one whom we need to talk to before talking to the child so that we can help the child." (CR, Zaka). This facilitated the CATS' focus on difficult household relational dynamics. At times activities needed to be aimed at the caregiver more than the adolescent, at other times they involved improving the relationship between the caregiver and the adolescent.

## Critical role of ongoing support and supervision for CATS

The CATS described the cumulative stress they felt from listening to the painful experiences of their clients, as well as trying to manage the impact of poor outcomes including clients passing away, "If a child tells you their problems, we need support so that we are not disturbed." (FGD, Beitbridge) They emphasised the critical importance of receiving psychosocial support through ongoing supervision from their mentors to cope. The CATS explained that they needed "some help here and there" to enable them to keep "working well when helping others" (CR, Kwekwe),

Supervision needed to intensify, through the provision of enhanced support and expertise, for cases involving more significant mental health problems, "I need you [the supervisor] to help me with ideas and skills on how to deal with very complicated problems [we discuss] during sessions." (CR, Chiredzi)

Although the majority of their direct needs were fulfilled by their mentors, the CATS also benefitted from being able to call on other professional adults, including nurses, primary counsellors and social workers to collaborate in providing and reinforcing discussions and intervention:

> He (client) wanted peace at home so that it would not disturb him in taking medication because when the fights started, he would run away without taking medication. The solution was to ask the parents to come to the hospital and undergo counselling from with a vice matron. (CR, Zaka)

## Discussion

Meeting the mental health needs of ALHIV is a critical aspect of providing care to this vulnerable population. Peer supporters can be an effective cadre to helping improve the health of their peers [16, 18]. However, as demonstrated in this study and elsewhere [17] the very things that make a peer supporter effective, including an understanding of the challenges of being a child or adolescent living with HIV based on lived experience, at the same time put them at emotional risk when they engage in the provision of care and support. They, too, face the problems and vulnerabilities related to living with HIV: including stigma and discrimination, orphanhood, poverty, medical problems and so on. Additionally, the complexity and relationally embedded nature of many of the problems encountered largely rest in the social environment as opposed to being internal to the ALHIV. This renders them not necessarily 'fixable'. The CATS can instead seek to ameliorate the harms of these structural problems.

As the CATS described their experiences of engaging with the complex problems faced by their peers, we were able to identify what support needs they had to be able to do the work effectively and without harm to themselves. We highlight the key areas that need operational attention if we are to responsibly shift first-line care to this cadre of young people. Attending to the lessons identified within this qualitative study, the authors developed a framework of principles to address the needs of young peer-supporters, TRUST (training, referral pathways,

understanding the remit of their role, supervision and mentorship, talking) to guide program development and service delivery.

## Training

First, they need training that provides them with a basic understanding of mental health: what promotes wellbeing and resilience and what contributes to the development of problems. Articulating these issues is important for understanding others, but it also provides information for understanding and validating their own experiences [29]. This study clearly demonstrated that peer supporters need guidance on how to address the problems they hear, spanning a range of severe and traumatic life experiences and circumstances. They need to be provided with a framework for approaching these problems, as some will be 'fixable' and others possible to shift and ameliorate, with harm reduction being the achievable outcome. It is critical for them to understand that their clients and they themselves won't be able to fully resolve many of the problems encountered, which are driven by poverty and structural inequity. They must not inadvertently feel that they failed because they did not achieve the unachievable, a consistent theme across work with the CATS [19]. Rather, they need to develop a greater understanding of the relational and structural complexity of what it is that they are discussing and trying to tackle [30].

## Referral pathways

The young peer supporter needs to work within the context of a larger system. There needs to be clear referral pathways to link them with other providers and services that go beyond what they themselves can offer. Some clients will need to be referred on for professional assessment, intervention or resources. These cases should first be discussed with the supervisor to ensure that appropriate referrals are made, and when needed, to help facilitate the referrals.

## Understanding the remit of their role

Along with referral pathways, there needs to be clearly defined parameters which delineate the scope of the peer supporters' role and the kinds of support they need to achieve this work. When expanding the role of peer support, we must recognize that peer supporters are not a direct substitute for mental health professionals in that they neither provide a diagnostic function nor treat mental illnesses. They are, however, well positioned to provide support for emotional wellbeing and assistance in coping with difficult challenges. In reality, this is not always as clear a distinction as it sounds and through supervision and case discussions peer supporters need specific guidance on decisions related to their role [31, 32].

## Supervision

Organizations that utilize young peer supporters have an ethical responsibility to support and protect these young people, whom are living with HIV themselves and may be affected by the pressures inherent in the peer supporter role [22]. This study builds on previous research that identified emotional risk to peer supporters and the pivotal role played by their mentors [20]. This need for support increases further when the role of the peer supporter expands to include mental health, which engages them more deeply in their own difficult experiences as well as their feelings of responsibility for alleviating the pain of others [31]. It is critical for supervisors to regularly provide peer supporters the opportunity to discuss any difficult feelings that are triggered, as well as to support them to moderate their expectations of themselves in terms of what they reasonably can and 'should' be able to accomplish regarding some of the 'unfixable'

problems of a client and the client's own choices. There should also be referral sources for mental health support in the event that the peer supporter's needs exceed what can be addressed in supervision.

Through mentorship and supervision, peer supporters need ongoing training that includes: articulation of their own experiences; further development of adaptive strategies to cope with the problems in their own lives; and how to practice self-care. Expanding their role to mental health problems and contributing factors, in order to serve confidently as a role model as they do in relation to their HIV specific challenges, requires that they first learn to talk about and address these issues in themselves. They need to experience the conversations addressing their challenging experiences, including their own potentially 'unsolvable' problems, their beliefs and thought patterns, in order to fully understand what is being asked of them.

## Talking

In Zimbabwe and throughout the region, depression and other mental health conditions are commonly stigmatised to varying extents [11] and young people are discouraged from talking about them [33]. Whether through using a formalized screening tool or general screening questions about mental health difficulties, peer supporters need specific wording for how to approach these conversations and what to ask about. These conversations break the silence, provide validation and an empathic ear and lay the groundwork for exploring options, coping strategies and strengthening support systems.

For each of the factors identified, Table 2, TRUST, outlines a set of guiding principles with core considerations to promote comprehensive preparation and ongoing support for peer supporters as they engage with the complex challenges faced by ALHIV.

## Recommendations for future research

This paper examined the experience of CATS as they provided care, including what the process was like for them and their assessment of what it was like for beneficiaries. Research is needed that looks at the experience and perspective of clients receiving support around psychological problems from young peer supporters in order to gain a full understanding of the benefits and risks of employing this cadre to provide mental health care. Additionally, research is needed to evaluate longer term outcomes in order to assess the efficacy of peer supporters in addressing problems challenging mental health. CATS appraised clients as benefiting from discussions of the problems in their lives based on direct feedback from clients as well as observation, seeing changes in mood and social interactions. It is not known if these changes are long lasting or more transient. Finally, it will be important to closely monitor implementation as peer supporters assume the responsibility of triage and direct support for mental health issues, ensuring adequate organizational structure to guide care, the provision of adequate support and mitigation of risk to young peer supporters. We propose that the TRUST framework can be a constructive guide to implementing much-needed occupational support for young peer-supporters. Further research to refine and further develop the TRUST framework would be valuable.

## Conclusion

Extending peer support to explicitly include a focus on mental health has enormous potential for improving the wellbeing of ALHIV and ultimately, improved virologic suppression. We illustrate that the training and support that is needed not only requires peer supporters to learn new information, but also to learn to use themselves in a new way. In bringing young peer supporters into this arena, we have a responsibility to set them up for success and to

**Table 2. TRUST: Guiding principles for expanding the role of peer supporters into mental health support, core considerations.**

| Recommendations | Core considerations for peer supporters |
|---|---|
| 1. **T**raining | Peer supporters need training to expand their knowledge base, build their skills and understanding that includes: |
| | • Information on common mental conditions: depression, anxiety, post-traumatic stress, substance abuse |
| | • Factors that contribute to psychological problems: disconnections, unresolved conflict, unaddressed loss |
| | • Complex, relationally and socially embeddedness of the problems contributing to the distress in ALHIV |
| | • Necessity of a systemic and preventive approach |
| | • Reconciling themselves to the fact that while they will be able to fix some things, there are others they will not |
| | • Successful impact may include amelioration of problems, where a harm reduction lens is adopted (increasing supports in order to live alongside some of the unchangeable problems). |
| | • While they can support and encourage a client, they do not have control over the choices made by the client or family. |
| | • Screening tool or clear guidelines on what to ask about and how |
| | • Guidance on how to explore options, decision making and intervention strategies |
| 2. **R**eferral pathways | • Clear referral pathways |
| | • Direction from supervisors on when to reach out for help and to whom, e.g. caregivers/ family members, supervisors, HCWs, other service providers |
| | • Professional assessment and intervention when distress is elevated to a degree that interferes with overall functioning, schooling, relationships |
| | • Services and resources such as those that address child protection issues, medical care, food security, schooling |
| | • Augment the effectiveness of this approach, sensitization of referral system about child and adolescent mental health needs |
| 3. **U**nderstanding the remit of their role | Peer supporters well positioned to: |
| | • Promote wellbeing and prevention of serious problems by reducing social and emotional isolation through ongoing engagement using basic counselling skills, (providing connection, empathy, validation, listening ear), as well as strengthening individual coping mechanisms and a social network |
| | • Reduce stressors and address factors that contribute to psychological problems: disconnections, unresolved conflict, unaddressed loss |
| | • Identify substantial needs that require referrals, e.g. child protection issues, serious psychological problems, tangible resources |
| 4. **S**upervision and mentorship | Supervision must appreciate complicating factors, respond to and support peer supporters in the following ways: |
| | • Potentially being triggered by engaging in conversations about complex problems of which they share experience |
| | • Limited experience of having previously talked about mental health problems |
| | • Increased risks from the cumulative effect of engaging more deeply with the nature of the complex problems and the emotional pain of others |
| | • Development of coping strategies, scaffolding and confidence to serve as role models related to mental health issues |
| | • Early support with sensitising caregivers to the peer supporters' role |
| | • Managing their expectations of realistic and feasible outcomes |
| | • Recognition of the value of engaging in self-care. |

(*Continued*)

**Table 2.** (Continued)

| Recommendations | Core considerations for peer supporters |
|---|---|
| 5. **T**alking helps | Peer supporters must be helped to: |
| | • Understand the power of talking when there is an empathic, listening ear |
| | • Develop a language to talk about the complexity of the problems they potentially face and |
| | • Have conversations about their own difficult experiences, both for support and to understand what these conversations look like. |

prevent potential emotional and psychological harms. To maximise the opportunities and mitigate the risks, we have drawn on our learning to identify a framework (TRUST) for guiding the implementation of peer supporter delivered mental healthcare for ALHIV. Adolescents need someone they trust that they can talk to and peer supporters need to trust they will be able to help and will themselves be okay in providing that support.

## Acknowledgments

We thank the CATS, the Zvandiri Mentors and the Zvandiri Interns and the study participants. We also appreciate the partnership with the Ministry of Health and Child Care.

## Author Contributions

**Conceptualization:** Carol Wogrin, Nicola Willis, Abigail Mutsinze, Dixon Chibanda.

**Data curation:** Abigail Mutsinze.

**Formal analysis:** Carol Wogrin, Abigail Mutsinze, Silindweyinkosi Chinoda, Sarah Bernays.

**Funding acquisition:** Nicola Willis.

**Methodology:** Carol Wogrin, Nicola Willis, Abigail Mutsinze.

**Project administration:** Nicola Willis, Ruth Verhey, Dixon Chibanda.

**Supervision:** Abigail Mutsinze, Silindweyinkosi Chinoda, Ruth Verhey.

**Writing – original draft:** Carol Wogrin, Sarah Bernays.

**Writing – review & editing:** Carol Wogrin, Nicola Willis, Abigail Mutsinze, Silindweyinkosi Chinoda, Ruth Verhey, Dixon Chibanda, Sarah Bernays.

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
