## [Decision Letter · Decision Letter 0]

14 Dec 2020

PONE-D-20-33576

It helps to talk: developing a guiding framework (TRUST) for peer support in delivering mental health care for adolescents living with HIV

PLOS ONE

Dear Dr. Wogrin,

Thank you for submitting your manuscript to PLOS ONE. After careful consideration, we feel that it has merit but does not fully meet PLOS ONE’s publication criteria as it currently stands. Therefore, we invite you to submit a revised version of the manuscript that addresses the points raised during the review process.

Please notice that the academic editor and both peer reviewers found difficult to follow the development of the TRUST framework. Please do elaborate and walk the readers throughout the process, from the Introduction to Methods to Results to Discussion. In qualitative research manuscripts and reports the distinction between interpreted Results/Findings and Discussion can be blurred. This is OK, but in this particular case we need to know whether the developed framework was only grounded on the study data or other sources and existing frameworks. 

We look forward to receiving your revised manuscript.

Kind regards,

Petros Isaakidis MD PhD

Academic Editor

PLOS ONE

Journal Requirements:

2.We note that you have indicated that data from this study are available upon request. PLOS only allows data to be available upon request if there are legal or ethical restrictions on sharing data publicly. For more information on unacceptable data access restrictions, please see http://journals.plos.org/plosone/s/data-availability#loc-unacceptable-data-access-restrictions.

Reviewers' comments:

Reviewer's Responses to Questions

**Comments to the Author**

1. Is the manuscript technically sound, and do the data support the conclusions?

Reviewer #1: Yes

Reviewer #2: Yes

2. Has the statistical analysis been performed appropriately and rigorously? 

Reviewer #1: Yes

Reviewer #2: Yes

3. Have the authors made all data underlying the findings in their manuscript fully available?

Reviewer #1: Yes

Reviewer #2: No

4. Is the manuscript presented in an intelligible fashion and written in standard English?

Reviewer #1: Yes

Reviewer #2: Yes

5. Review Comments to the Author

Reviewer #1: Dear Editor,

Thank you for trusting me to review this piece of work; It helps to talk: developing a guiding framework (TRUST) for peer support in delivering mental health care for adolescents living with HIV

I have read the manuscript and it is generally well written from the abstract to the conclusion with very few typos which I have pointed out

However, what was unclear to me is how the TRUST framework was developed; was it as a result of the data collected? For instance, I expected to see in the introduction something about the TRUST guiding framework which seems to be the crux of this paper but it is not mentioned until in the discussion. As a result, I found a disconnect between the results and discussion section. Once this is fixed, the manuscript will read better

Below are very few comments by section.

The Abstract was well written explaining the background up to conclusion.

Introduction was clear and understood. please remove the word "being in line 67" the sentence should read "social consequences of having an HIV positive status"

Methods used were well stated. Minor correction in line 145, remove the repeated word "then"

Results section

it would be good to summarize the themes before writing about them.

use different headings/ italics to differentiate themes and sub themes

in line 206, what structural support was there for CATs who felt overwhelmed, please elaborate

Discussion

I don't see a link between the results and discussion. you introduce the TRUST concepts which are not mentioned before at all. I would like to see a link between the two

Reviewer #2: Peer support models have been shown to improve retention and HIV viral load suppression among people living with HIV. The authors explored support needed during implementation of peer support for mental health services and propose guiding framework for the support needed using TRUST model.

Manuscript is well written. However has TRUST framework been tested/validated? What other models exist? These should be referenced and discussed in the manuscript.

The manuscript has some typographical errors that need to be corrected- line 35 "In sub-Sharan Africa", line 67 "social consequences of being having an"

Best regards

6. PLOS authors have the option to publish the peer review history of their article (what does this mean?). If published, this will include your full peer review and any attached files.

Reviewer #1: No

Reviewer #2: No

---

## [Author Response · Author response to Decision Letter 0]

15 Feb 2021

Dear Editor and Reviewers, 

Thank you for your thoughtful and helpful feedback on our paper. We appreciate the opportunity to respond and further strengthen our paper. 

We have done our best to respond to the comments – indeed we found the thoughtful reviews extremely helpful in our attempt to make the presentation of our thinking clearer. Thank you. 

We set out our response to the comments below. We refer to the line numbers in the track change version when directing the reviewers to our changes. We included the reviewers’ comments in bold italics and have copied and pasted the adjusted text into our response. 

We trust this is satisfactory as an approach.

Thank you.

RESPONSE TO COMMENTS FROM THE EDITORS AND REVIEWERS:

3. Have the authors made all data underlying the findings in their manuscript fully available?

Reviewer #1: yes, Reviewer #2: No

RESPONSE: There are ethical issues that restrict the sharing of the data that comprise the findings underlying this paper. The data is from case reviews and interviews conducted with adolescents and young people regarding their experiences and interventions providing metal health support with their HIV positive peers. The interviews contain potentially identifying information on the young peer counsellors as well as the peers they were supporting. Requests for data may be directed to: 

Nicola Willis, Executive Director

Africaid, 11-12 Stoneridge Way North, Avondale, Harare, Zimbabwe

Nicola@zvandiri.org

+263 4 335 805 

5. Review Comments to the Author

Reviewer #1: 

However, what was unclear to me is how the TRUST framework was developed; was it as a result of the data collected? For instance, I expected to see in the introduction something about the TRUST guiding framework which seems to be the crux of this paper but it is not mentioned until in the discussion. As a result, I found a disconnect between the results and discussion section. 

RESPONSE: Thank you for this observation. The TRUST framework was developed from our analysis of our own data, rather than a framework that had previously existed. To better describe the way in which to TRUST framework was developed so that its development is clear we have added the following text to the Introduction, Methods and Discussion sections. We believe these additions clarify the issue of the way in which the TRUST framework was developed, as well as better link the sections of the paper. 

We have added the following text to the Introduction (lines 111 – 114): 

We explore the perspectives and needs of young peer supporters as described by themselves when they take on the complexities of expanding their role to addressing the difficult life experiences that challenge their mental health of their peers. Based on our analysis of their accounts, we develop a framework outlining the overarching support and training principles (TRUST) that should be provided to guide the expansion of the peer supporter’s role in delivering critically needed mental health care to ALHIV. 

Added to the Methods section (lines 151 – 155): 

Analytical memos and weekly analytical team meetings were also used in the development of themes and identification of patterns related to CATS experiences across the datasets [28]. To maximise the value of this thematic analysis, we organised the key implications from our analysis into a framework of guiding operational principles to ensure that CATS are adequately supported to conduct mental health support on a sustained basis.

Discussion section: We have added two paragraphs to the start of the discussion section, with the second specifically addressing the development of the TRUST framework (lines 307– 313). 

As the CATS described their experiences of engaging with the complex problems faced by their peers, we were able to identify what support needs they had to be able to do the work effectively and without harm to themselves. We highlight the key areas that need operational attention if we are to responsibly shift first-line care to this cadre of young people. Attending to the lessons identified within this qualitative study, the authors developed a framework of principles to address the needs of young peer-supporters, TRUST (training, referral pathways, understanding the remit of their role, supervision and mentorship, talking) to guide program development and service delivery. 

Introduction was clear and understood. Please remove the word "being in line 67" the sentence should read "social consequences of having an HIV positive status": 

RESPONSE: Done 

Methods used were well stated. Minor correction in line 145, remove the repeated word "then": 

RESPONSE: Done 

Results section it would be good to summarize the themes before writing about them. 

RESPONSE: Thank you for this recommendation. We agree that providing a summary of the themes at the start of the section strengthens the results section and we have added the following (lines 159 - 167): 

In the CATS’ accounts of supporting their peers who were dealing with depression and anxiety, participants described a number of challenges and opportunities in their experiences. We identified three overarching themes. These were: the process, described as the journey, of learning to talk about mental health problems; support structures and strategies for the CATS to deliver sustained care; and the critical role of support and supervision. The journey, learning to talk about mental health problems further broke down into four sub-themes: strengths of peer supporters, challenges in ‘moving into mental health support, the complexity and relational embeddedness of the problems faced by ALHIV, and making it ‘less bad’: revised definitions of success. 

use different headings/ italics to differentiate themes and sub themes: 

RESPONSE: We have changed the formatting so that it follows the PLOS ONE guidelines. 

in line 206, what structural support was there for CATs who felt overwhelmed, please elaborate 

RESPONSE: We have added the following to better describe the approach taken to support the CATS when they were overwhelmed (lines 207 – 208):

This also predisposed them to be triggered when engaging with similar experiences, and for which they needed additional support through discussion of these issues with their supervisors and mentors.

Additionally, we further emphasised this issue in the discussion. The underlined text denotes the additions that we have made to the existing text (lines 352 – 357): ‘It is critical for supervisors to regularly provide peer supporters the opportunity to discuss any difficult feelings that are triggered, as well as to support them to moderate their expectations of themselves in terms of what they reasonably can and ‘should’ be able to accomplish regarding some of the ‘unfixable’ problems of a client and the client’s own choices. There should also be referral sources for mental health support in the event that the peer supporter’s needs exceed what can be addressed in supervision. 

I don't see a link between the results and discussion. you introduce the TRUST concepts which are not mentioned before at all. I would like to see a link between the two

RESPONSE: Thank you for this recommendation. We have added two paragraphs to the beginning of the discussion section. We believe that this now explicitly links the results and discussion and provides better context for the TRUST framework. The paragraphs added are as follows (lines 295 – 313): 

Meeting the mental health needs of ALHIV is a critical aspect of providing care to this vulnerable population. Peer supporters can be an effective cadre to helping improve the health of their peers [16,18]. However, as demonstrated in this study and elsewhere [17] the very things that make a peer supporter effective, including an understanding of the challenges of being an child or adolescent living with HIV based on lived experience, at the same time put them at emotional risk when they engage in the provision of care and support. They, too, face the problems and vulnerabilities related to living with HIV, including stigma and discrimination, orphanhood, poverty, medical problems and so on. Additionally, the complexity and relationally embedded nature of many of the problems encountered largely rest in the social environment as opposed to being internal to the ALHIV. This renders them not necessarily ‘fixable’. The CATS can instead seek to ameliorate the harms of these structural problems. 

As the CATS described their experiences of engaging with the complex problems faced by their peers, we were able to identify what support needs they had to be able to do the work effectively and without harm to themselves. We highlight the key areas that need operational attention if we are to responsibly shift first-line care to this cadre of young people. Attending to the lessons identified within this qualitative study, the authors developed a framework of principles to address the needs of young peer-supporters, TRUST (training, referral pathways, understanding the remit of their role, supervision and mentorship, talking) to guide program development and service delivery. 

Reviewer #2

TRUST framework been tested/validated? What other models exist? These should be referenced and discussed in the manuscript.

RESPONSE: The TRUST framework emerged from analysis of the findings. As they are a novel output from this study and manuscript, there has not been an opportunity for these guidelines to have been validated yet. To the authors’ knowledge, this examination of the relatively new approach of expanding the role of AYPLHIV peer supporters beyond psychosocial support and into the area of being first-line responders to more complex mental health issues is new and has received very little attention. The shift towards considering youth peer-supporters to provide mental health support in resource-constrained settings though is gaining considerable traction from funders and policy makers, as described in the Introduction. We are not aware of other models that specifically examine and outline the needs of the young peer supporter as they assume these challenging responsibilities, hence the need to develop guidelines. 

The manuscript has some typographical errors that need to be corrected- line 35 "In sub-Sharan Africa", line 67 "social consequences of being having an": 

RESPONSE: Done

---

## [Editor Report · Decision Letter 1]

18 Feb 2021

It helps to talk: a guiding framework (TRUST) for peer support in delivering mental health care for adolescents living with HIV

PONE-D-20-33576R1

Dear Dr. Wogrin,

We’re pleased to inform you that your manuscript has been judged scientifically suitable for publication and will be formally accepted for publication once it meets all outstanding technical requirements.

Kind regards,

Petros Isaakidis

Academic Editor

PLOS ONE
---

## [Editor Report · Acceptance letter]

22 Feb 2021

PONE-D-20-33576R1 

It helps to talk: a guiding framework (TRUST) for peer support in delivering mental health care for adolescents living with HIV 

Dear Dr. Wogrin:

I'm pleased to inform you that your manuscript has been deemed suitable for publication in PLOS ONE. Congratulations! Your manuscript is now with our production department. 

Kind regards, 

on behalf of

Dr. Petros Isaakidis 

Academic Editor

PLOS ONE